# Organophosphate Flame Retardants and Perfluoroalkyl Substances in Drinking Water Treatment Plants from Korea: Occurrence and Human Exposure

**DOI:** 10.3390/ijerph18052645

**Published:** 2021-03-05

**Authors:** Wonjin Sim, Sol Choi, Gyojin Choo, Mihee Yang, Ju-Hyun Park, Jeong-Eun Oh

**Affiliations:** 1Education & Research Center for Infrastructure of Smart Ocean City (i-SOC Center), Pusan National University, Busan 46241, Korea; wjsim81@gmail.com; 2Department of Civil and Environmental Engineering, Pusan National University, Busan 46241, Korea; cs34193059@gmail.com (S.C.); gjchoo@korea.kr (G.C.); 3National Fishery Products Quality Management Service, Busan 48943, Korea; 4Department of Environmental Infrastructure Research, National Institute of Environmental Research, Ministry of Environment, Incheon 22689, Korea; mhyang@korea.kr (M.Y.); soyang@korea.kr (J.-H.P.)

**Keywords:** organophosphate flame retardants (OPFR), perfluoroalkyl substances (PFAS), drinking water treatment plants, occurrence, human exposure

## Abstract

In this study, the concentrations of organophosphate flame retardants (OPFR) and perfluoroalkyl substances (PFAS) were investigated in raw water and treated water samples obtained from 18 drinking water treatment plants (DWTPs). The ∑_13_OPFR concentrations in the treated water samples (29.5–122 ng/L; median 47.5 ng/L) were lower than those in the raw water (37.7–231 ng/L; median 98.1 ng/L), which indicated the positive removal rates (0–80%) of ∑_13_OPFR in the DWTPs. The removal efficiencies of ∑_27_PFAS in the DWTPs ranged from −200% to 50%, with the ∑_27_PFAS concentrations in the raw water (4.15–154 ng/L; median 32.0 ng/L) being similar to or lower than those in the treated water (4.74–116 ng/L; median 42.2 ng/L). Among OPFR, tris(chloroisopropyl) phosphate (TCIPP) and tris(2-chloroethyl) phosphate (TCEP) were dominant in both raw water and treated water samples obtained from the DWTPs. The dominant PFAS (perfluorooctanoic acid (PFOA) and perfluorohexanoic acid (PFHxA)) in the raw water samples were slightly different from those in the treated water samples (PFOA, L-perfluorohexane sulfonate (L-PFHxS), and PFHxA). The 95-percentile daily intakes of ∑_13_OPFR and ∑_27_PFAS via drinking water consumption were estimated to be up to 4.9 ng/kg/d and 0.22 ng/kg/d, respectively. The hazard index values of OPFR and PFAS were lower than 1, suggesting the risks less than known hazardous levels.

## 1. Introduction

Organophosphate flame retardants (OPFR) and perfluoroalkyl substances (PFAS) are used widely in various industrial and commercial applications [1,2,3]. Because OPFR are considered alternatives to halogenated flame retardants, their share has increased to 20% of the total use of flame retardants in Europe [4]. PFAS can offer resistance against water, oil, and soil owing to their structures with both hydrophobic and hydrophilic functional groups [5]. Therefore, they are used as surface protectors and surfactants of carpets, leathers, textiles, papers, and fire extinguishing agents. The manufacture and use of products containing OPFR and PFAS release their traces into the environment [4,6]. The exposure to OPFR is known to have adverse effects, such as inflammation of the eyes and ears, carcinogenicity, dermatitis, neurotoxicity, and physical development disorder, on human health [4,7,8]. PFAS have been classified as potentially carcinogenic substances by the US Environmental Protection Agency (EPA) and the Organization for Economic Cooperation and Development (OECD) [9].

Globally, OPFR and PFAS have been regulated owing to their adverse effects on human health and the ecosystem [10,11,12]. The use of tris(2-chloroethyl) phosphate (TCEP), tris(chloroisopropyl) phosphate (TCIPP), and tris(1,3-dichloro-2-propyl) phosphate (TDCIPP) in children’s products has been banned [13,14,15]. The use of perfluorooctane sulfonate (PFOS), perfluorooctanoic acid (PFOA), and related substances has been restricted under the Stockholm Convention for Persistent Organic Pollutants and the European Union’s Regulation on Registration, Evaluation, Authorization, and Restriction of Chemicals [16,17,18]. Despite these regulations on the use of OPFR and PFAS, these chemicals have often been detected in water resources because of their widespread use [19,20]. However, it is difficult to remove OPFR and PFAS using conventional water treatment processes [21,22]. Therefore, OPFR and PFAS have received significant attention as emerging pollutants in drinking water, and many studies have been performed to investigate their concentrations in drinking water and risks they pose to human health [19,23,24,25].

Several studies have been reported on OPFR and PFAS in drinking water in South Korea. Heo et al. (2014) investigated 16 PFAS in foods, including tap water; Lee et al. (2016) reported concentrations of 10 OPFR in tap water, purified water, and bottled water [19,26]; and Park et al. (2018) performed a survey on nine OPFR and 14 PFAS in tap water samples [24]. Using raw water and treated water from drinking water treatment plants (DWTPs), Choo and Oh (2020) investigated nine OPFR in six sites; and Kim et al. (2020) reported concentrations of 14 PFAS at five DWTPs [27,28]. In South Korea, DWTPs mostly treat surface water (rivers, lakes, and reservoirs). Surface water sources can be easily contaminated by various hazardous substances discharged from industrial facilities and cities located along and around them [29]. Therefore, the fate properties of pollutants such as OPFR and PFAS must be investigated in DWTPs for the safety of drinking water. However, only a few studies have recently been performed to determine the concentrations of OPFR or PFAS in raw water and treated water obtained from DWTPs [27,28]. Therefore, the monitoring of OPFR and PFAS is necessary in various full-scale DWTPs to investigate their occurrence and removal.

In this study, 13 OPFR and 27 PFAS were determined in raw water and treated water samples collected from 18 full-scale DWTPs located from upstream to downstream of the Nakdong River basin. In particular, two OPFR (tripropyl phosphate (TPrP) and diphenyl cresyl phosphate (DCP)) and ten PFAS (perfluoropentane sulfonate (PFPeS), perfluorononane sulfonate (L-PFNS), five precursors, and three alternatives) were first reported in both raw water and treated water samples obtained from the DWTPs in South Korea. The distribution and removal of OPFR and PFAS in full-scale DWTPs were also assessed based on their concentrations in the samples. In addition, the daily intakes of OPFR and PFAS via drinking water consumption were estimated.

## 2. Materials and Methods

### 2.1. Chemicals and Materials

13 OPFR and 27 PFAS were analyzed in this study. The target OPFR were triethyl phosphate (TEP), TCEP, TCIPP, TDCIPP, triphenyl phosphate (TPhP), tricresyl phosphate (TCP), tri-n-butyl phosphate (TNBP), TPrP, ethylhexyldiphenyl phosphate (EHDPP), DCP, tris(2-butoxyethyl) phosphate (TBOEP), tris(2-ethylhexyl) phosphate (TEHP), and tri-isobutyl phosphate (TIBP). The target PFAS were 10 perfluorinated carboxylic acids (PFCAs) (perfluoropentanoic acid (PFPeA), perfluorohexanoic acid (PFHxA), perfluoroheptanoic acid (PFHpA), PFOA, perfluorononanoic acid (PFNA), perfluorodecanoic acid (PFDA), perfluoroundecanoic acid (PFUnDA), perfluorododecanoic acid (PFDoDA), perfluorotridecanoic acid (PFTrDA), and perfluorotetradecanoic acid (PFTeDA)), seven linear isomers of perfluoroalkyl sulfonic acids (PFSAs) (perfluorobutane sulfonate (L-PFBS), L-PFPeS, perfluorohexane sulfonate (L-PFHxS), perfluoroheptane sulfonate (L-PFHpS), L-PFOS, L-PFNS, and perfluorodecane sulfonate (L-PFDS)), two branched isomers of PFSAs (B-PFHxS and B-PFOS), five precursors (N-methyl perfluorooctane sulfonamidoacetic acid (N-MeFOSAA), N-ethyl perfluorooctane sulfonamidoacetic acid (N-EtFOSAA), 1H,1H,2H,2H-perfluorohexane sulfonate (4:2FTS), 1H,1H,2H,2H-perfluorooctane sulfonate (6:2FTS), and 1H,1H,2H,2H-perfluorodecane sulfonate (8:2FTS)), and three alternatives (3H-perfluoro-3-(3-methoxy-propoxy)propanoic acid (ADONA), hexafluoropropylene oxide dimer acid (GenX; HFPO-DA), and 9-chlorohexadecafluoro-3-oxanonane-1-sulfonate (9Cl-PF3ONS; major component of F-53B)). The details of chemicals and materials are given in the Appendix A.

### 2.2. Sample Collection

The raw water (*n* = 36) and treated water (drinking water; *n* = 36) samples were collected from 18 DWTPs along the Nakdong River in South Korea. The sites were divided into the upstream (DWTPs 1–5), midstream (DWTPs 6–10), and downstream (DWTPs 11–18) depending on the locations. The source of raw water for 15 DWTPs was surface river water, whereas the remaining three DWPTs received the raw water supplies from the riverbed water or riverbank filtration. The samples were obtained in two different months: in August and October 2019. Conventional DWTPs generally use coagulation, sedimentation, rapid sand filtration, and chlorination processes to purify the raw water. However, in this study, all the DWTPs, except one site, had additional advanced drinking water treatment processes such as ozonation and/or granular activated carbon (GAC). The details of each sampling site are given in the Appendix A. All water samples were placed in pre-cleaned amber glass bottles (2 L), transported to the laboratory using ice-boxes at 4 °C, and stored below −20 °C for further treatment and analysis.

### 2.3. Sample Preparation

For the analysis of OPFR, 500 mL of water sample was spiked with 10 ng of internal standards and extracted by solid phase extraction (SPE) using an Oasis HLB cartridge (200 mg, 6 cm^3^; Waters, Milford, MA, USA) preconditioned with a sequence of 6 mL of dichloromethane (DCM), 6 mL of methanol (MeOH), and 6 mL of water. After sample loading, the cartridge was dried for 30 min using vacuum and eluted with 8 mL of DCM. The extract was concentrated to 100 µL under a gentle stream of nitrogen using a Turbo Vap II (Caliper Life Sciences, Hopkinton, MA, USA) and transferred to an amber vial with an injection of syringe standard (10 ng).

For the determination of PFAS, 500 mL of water sample was spiked with 5 ng of internal standards after filtration with a glass fiber filter (GF/F; Whatman, Maidstone, UK). The SPE cartridge (Oasis WAX, 200 mg, 6 cm^3^; Waters) was preconditioned with a sequence of 6 mL of basic MeOH (0.1% ammonia), 6 mL of MeOH, and 6 mL of water. After sample loading, the cartridge was washed with 6 mL of 25 mM ammonium acetate and dried for 30 min using vacuum, and finally eluted with 4 mL of basic MeOH and 4 mL of MeOH. The extract was concentrated to 500 µL under a gentle stream of nitrogen using a Turbo Vap II, 500 µL of MeOH was added, and the syringe standards (5 ng) were added prior to analysis.

### 2.4. Instrumental Analysis

An Agilent 7890B gas chromatography coupled with an Agilent 7000C tandem mass spectroscopy (Agilent Technologies, Santa Clara, CA, USA) was used for the analysis of OPFR. A DB-5MS UI column (15-m long, 0.25-mm i.d., 0.10-μm film thickness; J&W Scientific, Palo Alto, CA, USA) was used to separate the OPFR in the extracts. The oven program was 50 °C for 3 min, increased by 15 °C/min to 230 °C, and then by 15 °C/min to 300 °C (held for 1 min). Helium was used as the carrier gas with a constant flow at 1.5 mL/min. Inlet, interface, and source temperatures were maintained at 300 °C, 280 °C, and 300 °C, respectively. Selected reaction monitoring (SRM) and positive electron ionization modes were used with 70 eV ionization voltage.

The identification of PFAS was performed by an Agilent 1200 high-performance liquid chromatography (HPLC) system coupled with an Agilent 6460 electrospray triple-quadruple mass spectrometer (ESI-MS-MS). A guard column (4.6-mm i.d., 12.5-mm length, 5-μm particles; Agilent) and a ZORBAX Eclipse XDB-C18 HPLC column (4.6-mm i.d., 150-mm length, 3.5-μm particles; Agilent) were used to separate the PFAS from the complex matrices present in the extracts. A binary mobile phase (MeOH and 2 mM ammonium acetate in water) gradient was used, and the target compounds were identified and quantified using SRM. The details of instrumental analysis conditions are shown in the Appendix A.

### 2.5. Quality Assurance/Qaulity Control

The concentrations of the target compounds in the samples were within the calibration range (1–500 ng/mL for OPFR; 0.05–50 ng/mL for PFAS), and the correlation coefficients (r^2^) of the calibration curves were higher than 0.99 for all curves. The relative standard deviations of the relative response factors of each compound in the calibration solutions were all below ±15%. Due to the widespread use of OPFR and PFAS in laboratory products, such as clothing, tubes, cartridges, and gloves, all of the products used to analyze the OPFR and PFAS were pre-cleaned with acetonitrile three times to prevent potential sample contamination. Field blanks were placed in sample containers that underwent identical handling, shipment, and analysis as the field water samples, and procedural blank samples were also included in every batch (10–15 samples). The concentrations of the field and procedural blanks were always lower than the method detection limits (MDLs) of all target compounds. The MDLs were defined as three times the standard deviation of the measured concentration in seven replicated water samples spiked target compounds, which ranged from 0.23–1.10 ng/L for OPFR, and from 0.20–1.09 ng/L for PFAS (Appendix A in the Appendix A). The accuracies (OPFR: 104 ± 9%, PFAS: 88 ± 11%) and precisions (<20%) were obtained from water samples (*n* = 3) spiking OPFR (10 ng of native and internal standards) and PFAS (2.5 ng of native standards and 5 ng of internal standards). The recoveries of the mass-labeled internal standards in water samples were within the acceptable ranges for all samples: 91 ± 14% for TCEP-d_12_, 79 ± 15% for TCIPP-d_18_, 93 ± 13% for TDCIPP-d_15_, 84 ± 23% for TPhP-d_15_, 82 ± 8.3% for MPFHxA, 98 ± 18% for MPFHxS, 98 ± 6.7% for MPFOA, 96 ± 18% for MPFOS, 100 ± 11% for MPFNA, 92 ± 13% for MPFDA, 77 ± 14% for MPFUnDA, and 70 ± 18% for MPFDoDA.

### 2.6. Calculation of Estimated Daily Intake

The daily intakes of OPFR and PFAS via drinking water consumption were estimated using an exposure assessment formula developed by the US Environmental Protection Agency [30]. The formula considers the observed OPFR and PFAS concentrations in the treated water (C; ng/L), the daily consumption rate of drinking water (R; L/d), the body weight (BW; kg), the estimated daily intake (EDI; ng/kg/d), the oral reference dose (RfD; ng/kg/d) and/or the tolerable daily intake (TDI; ng/kg/d), the hazard quotient (HQ), and the hazard index (HI). The R values per body weight (R/BW) were obtained from the Korean Exposure Factors Handbooks for adults and children [31,32]. The EDI values were calculated using the Monte Carlo simulations (100,000 iterations) to decrease any uncertainties in the risk assessment. The RfD and/or TDI values for TNBP, TCEP, TCIPP, TBOEP, TPhP, PFOA, and PFOS were 2400, 2200, 8000, 1500, 7000, 1500, and 150 ng/kg/d respectively [5,27]. The potential non-cancer risks posed by OPFR and PFAS in the drinking water were assessed by comparing the EDI values with the RfDs and/or TDIs.
Estimated daily intake (EDI) = C × R ÷ BW(1)
Hazard quotient (HQ) = EDI ÷ RfD (or TDI)(2)
Hazard index (HI) = ∑HQs (sum of HQs)(3)

## 3. Results and Discussion

### 3.1. Concentrations of OPFR and PFAS in Raw Water and Treated Water

Figure 1 depicts the concentrations of ∑_13_OPFR and ∑_27_PFAS in the raw water and treated water samples obtained from the DWTPs. The Mann–Whitney U test was performed using SPSS 25.0 (IBM Corp., Armonk, NY, USA) to compare the concentrations of the two groups. The concentrations below MDL (not detected; ND) were assigned as zero value for the statistical analysis. The detailed concentrations of OPFR and PFAS are summarized in the Appendix A. The raw water samples had significantly higher ∑_13_OPFR concentrations (37.7–231 ng/L; median 98.1 ng/L) than those in the treated water (29.5–122 ng/L; median 47.5 ng/L) (*p* < 0.05, Mann–Whitney U test). TCIPP (raw water: median 30.8 ng/L, treated water: median 16.2 ng/L) and TCEP (raw water: median 26.9 ng/L, treated water: median 11.8 ng/L) were dominantly detected in the DWTPs, followed by TBOEP (raw water: median 12.1 ng/L, treated water: median 7.52 ng/L) and TEP (raw water: median 7.90 ng/L, treated water: median 2.93 ng/L). The concentrations of these dominant OPFR in the raw water were lower than or similar to those in the Nakdong River basin reported previously by Seo et al. (2015) (TCIPP: ND–519 ng/L, TCEP: ND–210 ng/L, TBOEP: ND–1865 ng/L), Choo et al. (2018) (TCIPP: 17.7–495 ng/L, TCEP: 15.0–234 ng/L, TBOEP: 11.4–156 ng/L), and Choo and Oh (2020) (TCIPP: 7.09–109 ng/L, TCEP: 9.89–68.9 ng/L, TBOEP: 7.56–52.8 ng/L) [27,33,34]. The concentrations in the treated water samples were similar to or lower than the OPFR concentrations previously reported in South Korea (TCIPP: 7.05–108 ng/L, TCEP: 8.13–87.4 ng/L), China (TCIPP: 14.4–109 ng/L, TCEP: 28.5–139 ng/L), and USA (TCIPP: 210 ng/L, TCEP: 120 ng/L) [24,27,35,36].

Unlike the results of OPFR, the ∑_27_PFAS concentrations in the raw water (4.15–154 ng/L; median 32.0 ng/L) did not statistically differ from those in treated water (4.74–116 ng/L; median 42.2 ng/L) (*p* > 0.05, Mann–Whitney U test). In the raw water, PFOA (median 8.82 ng/L) and PFHxA (median 8.03 ng/L) were the dominant, whereas L-PFHxS (median 8.69 ng/L), PFOA (median 9.94 ng/L), and PFHxA (median 9.52 ng/L) had relative high concentrations in the treated water. The detection frequencies and concentrations of precursors and alternatives were lower than those of PFCAs and PFSAs. N-EtFOSAA and 6:2FTS were only detected in the samples among the precursors and alternatives. The occurrences of PFAS were similar to or lower than those in the surface water and drinking water in South Korea as reported by Kim et al. (2020) (PFOA: 1.5–65.2 ng/L, PFHxA: 1.4–33.7 ng/L, PFHxS: 0.5–600 ng/L) and Park et al. (2018) (PFOA: 0.755–27.7 ng/L, PFHxA: 0567–24.1 ng/L, PFHxS: 0.381–190 ng/L) [24,28]. The concentrations in the treated water samples were similar to or lower than the PFAS concentrations previously reported in China (PFOS: <0.1–14.8 ng/L, PFOA: <0.1–109 ng/L), Japan (PFOS: 0.03–50.9 ng/L, PFOA: 0.18–84 ng/L), USA (PFOS: <1.0–64 ng/L, PFOA: 1.2–7200 ng/L), and Spain (PFOS: <0.12–71 ng/L, PFOS: 0.32–57.4 ng/L) [20].

To investigate the temporal trends in OPFR and PFAS, their concentrations in water samples obtained from DWTPs along the Nakdong River between 2017 and 2019 were compared (Figure 2). The detailed concentrations of OPFR and PFAS in 2017 and 2018 are summarized in the Appendix A. The total concentrations of OPFR in 2017 (raw water: 39.1–245 ng/L; median 113 ng/L, treated water: 30.8–123 ng/L; median 62.6 ng/L) were similar to those found in 2019, and the dominant compounds were TCEP and TCIPP as in this study [27]. Unlike OPFR, the total concentrations of PFAS in 2019 were lower than the results in 2017 (raw water: 8.99–645 ng/L; median 93.8 ng/L, treated water: 6.3–493 ng/L; median 65.1 ng/L) and 2018 (treated water: 10–173 ng/L; median 106 ng/L) [28,37]. Especially, PFHxS was the most dominant compound in 2017 (raw water: 0.5–600 ng/L; median 42.9 ng/L, treated water: 0.5–454 ng/L; median 18.0 ng/L) and 2018 (treated water: ND–126 ng/L; median 48.5 ng/L) [28,37]. According to the 2018 survey, this was due to the discharge of PFHxS from industrial facility located in the upstream of the Nakdong River [37]. The decreases of PFAS concentrations in 2019 were related to the recently implemented drinking water guidelines in 2018 (480 ng/L for PFHxS; 70 ng/L each for PFOS and PFOA) owing to the PFAS accident [37].

Figure 3 shows the concentrations of ∑_13_OPFR and ∑_27_PFAS in the raw water according to locations of water sources. The ∑_27_PFAS concentrations in the raw water clearly indicated statistical differences based on the locations: upstream (4.15–30.9 ng/L; median 14.5 ng/L), midstream (37.0–154 ng/L; median 62.4 ng/L), and downstream (6.00–80.4 ng/L; median 32.0 ng/L) (*p* < 0.05, Mann–Whitney U test). This indicates that there are pollution sources of PFAS above the midstream of the Nakdong River. According to Kim et al. (2016), electronics and textile industrial wastewater treatment plants (WWTPs) had relatively high discharge loads of PFAS in South Korea. Notably, large electronics and textile industrial complexes are located above the midstream of the Nakdong River [38]. Therefore, the occurrence of PFAS might be affected by these industrial facilities. Unlike PFAS, the concentrations of ∑_13_OPFR in raw water did not vary statistically based on the locations: upstream (62.2–145 ng/L; median 88.3 ng/L), midstream (67.7–231 ng/L; median 105 ng/L), and downstream (37.7–131 ng/L; median 103 ng/L) (*p* > 0.05, Mann–Whitney U test). These results indicate that OPFR may have different sources from PFAS in the Nakdong River like the discharge of OPFR from extensive plastic use because they have been applied as additives of synthetic resins [1]. Park et al. (2018) and Lee et al. (2016) reported the possibility of OPFR contamination by plastic products [19,24].

### 3.2. Distribution and Remvoal of OPFR and PFAS in DWTPs

Figure 4 shows the distributions of OPFR and PFAS in the water samples obtained from the DWTPs. In the raw water, TCIPP (mean 33.3%) had the highest proportions among OPFR, followed by TCEP (27.7%) and TBOEP (13.8%). The distributions of OPFR in the treated water were similar to those in the raw water (TCIPP: 31.7%, TCEP: 29.0%, TBOEP: 16.4%). However, the distribution patterns of PFAS were slightly different between the raw water (PFOA: 25.4%, PFHxA: 24.4, PFHpA: 9.9%, L-PFHxS: 9.0%) and treated water (PFOA: 22.6%, L-PFHxS: 22.4%, PFHxA: 22.3%, PFHpA: 7.8%). The proportion of L-PFHxS significantly increased from the raw water to the treated water. These distribution patterns of OPFR and PFAS appeared to be related to their removal rates in the DWTPs. Table 1 lists the removal tendencies of OPFR and PFAS in the DWTPs. The removal rates of TCIPP, TCEP, and TBOEP mostly ranged from 0% to 80%. Because these substances accounted for a significant proportion of OPFR, their distribution patterns in the raw water and treated water samples were similar. The removal rates of PFOA, PFHxA, and PFHpA mostly ranged from −100% to 50%, whereas the L-PFHxS removal rates were less than −200% in more than half of the cases. Therefore, the proportions of L-PFHxS in the treated water were higher than those in the raw water.

The removal characteristics of OPFR in DWTPs or WWTPs were reported to be slightly different based on the process types. Several studies have reported that TCIPP and TCEP have poor removal rates in conventional DWTPs or WWTPs because of their solubilities and formation from precursors during treatment processes [27,39,40]. However, OPFR indicated positive removal efficiencies (14–92%) in advanced treatment processes such as ozonation and GAC in full-scale DWTPs [27]. In this study, OPFR were mostly removed by the DWTPs that had the advanced treatment processes such as ozonation and/or GAC. PFAS were not effectively removed by ozonation and chlorination processes, whereas the GAC processes had high removal efficiencies for PFAS than those in the other water treatment processes [20,28,41]. However, the removal rates of PFAS may decrease because breakthrough is frequently caused by the low sorption capacity of PFAS in GAC processes [20,28]. Therefore, the PFAS removal rates in DWTPs have been reported to vary from negative to positive in several surveys, including in this study [20,28,41]. Kim et al. (2020) reported that the removal efficiencies of PFAS increased as the perfluorocarbon chain length increased because of GAC treatment processes [28]. This study also indicated the PFAS removal patterns in DWTPs were similar to those reported by Kim et al. (2020) [28].

### 3.3. Exposure Assessment of OPFR and PFAS via Drinking Water Consumption

The EDIs of OPFR and PFAS were calculated through the Monte Carlo simulations using SPSS 25.0. The EDIs and potential non-cancer risks of OPFR and PFAS are summarized in the Appendix A. The frequency charts of the EDIs of OPFR and PFAS from the Monte Carlo simulations are given in the Appendix A. The 50-percentile EDIs of ∑_13_OPFR and ∑_27_PFAS were 1.9 ng/kg/d and 1.5 ng/kg/d, respectively, for toddlers (ages 1–2 years), followed by preschoolers (ages 3–6 years; ∑_13_OPFR: 1.7 ng/kg/d, ∑_27_PFAS: 1.3 ng/kg/d), school-aged-children (ages 7–12 years; ∑_13_OPFR: 1.2 ng/kg/d, ∑_27_PFAS: 0.93 ng/kg/d), adults (∑_13_OPFR: 0.85 ng/kg/d, ∑_27_PFAS: 0.66 ng/kg/d), and adolescents (ages 13–18 years; ∑_13_OPFR: 0.83 ng/kg/d, ∑_27_PFAS: 0.65 ng/kg/d). The 95-percentile EDIs of ∑_13_OPFR and ∑_27_PFAS were 4.9 ng/kg/d and 5.7 ng/kg/d, respectively, for toddlers (ages 1–2 years), followed by preschoolers (ages 3–6 years; ∑_13_OPFR: 4.2 ng/kg/d, ∑_27_PFAS: 4.7 ng/kg/d), school-aged-children (ages 7–12 years; ∑_13_OPFR: 3.0 ng/kg/d, ∑_27_PFAS: 3.6 ng/kg/d), adults (∑_13_OPFR: 2.2 ng/kg/d, ∑_27_PFAS: 2.5 ng/kg/d), and adolescents (ages 13–18 years; ∑_13_OPFR: 2.1 ng/kg/d, ∑_27_PFAS: 2.4 ng/kg/d). The EDIs of OPFR and PFAS for toddlers were the highest based on the daily consumption rate of drinking water per body weight. The 95-percentile HQs of TNBP, TCEP, TCIPP, TBOEP, TPhP, PFOA, and total PFOS ranged from 1.5 × 10^−5^ to 1.5 × 10^−3^. The 95 percentile HI values were 3.9 × 10^−3^ for toddlers, 3.3 × 10^−3^ for preschoolers, 2.4 × 10^−3^ for school-aged-children, 1.7 × 10^−3^ for adults, and 1.6 × 10^−3^ for adolescents, indicating the risks less than known hazardous levels (defined as HI < 1).

## 4. Conclusions

In this study, contamination of OPFR and PFAS were investigated in raw water and treated water samples obtained from the 18 DWTPs in South Korea. Among OPFR and PFAS monitored in this study, TCIPP, TCEP, PFOA, and PFHxA were dominant in both raw water and treated water samples. The occurrence patterns of OPFR and PFAS in raw water and treated water samples appeared to be affected by their removal rates in the DWTPs. The full-scale DWTPs exhibited positive removal rates for OPFR, whereas the PFAS removal rates varied widely, from negative to positive values. Therefore, the ∑_13_OPFR concentrations in the treated water were lower than those in the raw water, whereas the ∑_27_PFAS concentrations showed no statistical differences between the raw water and treated water. Nevertheless, the HI values of OPFR and PFAS via drinking water consumption were lower than 1, indicating the risks less than known hazardous levels. Because the concentrations of hazardous substances in drinking water depend on the water sources and the treatment processes, pollutants such as OPFR and PFAS must be removed effectively by DWTPs, considering the possible contamination of the water environment. However, as mentioned above, the concentrations of PFAS were often high in the treated water even after the GAC process. Therefore, further research is necessary to optimize water treatment processes for the safety of drinking water in South Korea.

## Figures and Tables

**Figure 1 ijerph-18-02645-f001:**
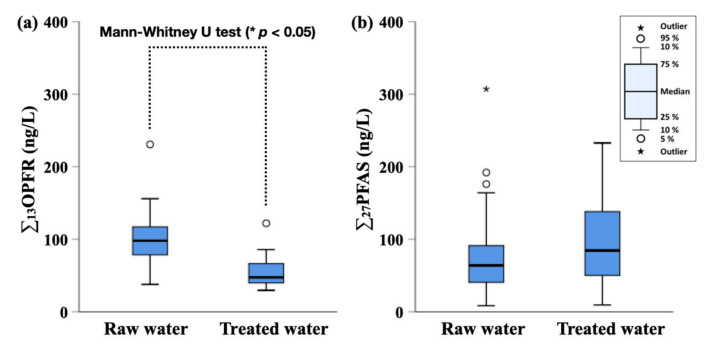
Comparison of ∑_13_OPFR and ∑_27_PFAS in raw water and treated water: (**a**) ∑_13_OPFR; (**b**) ∑_27_PFAS. OPFR: organophosphate flame retardants; PFAS: perfluoroalkyl substances.

**Figure 2 ijerph-18-02645-f002:**
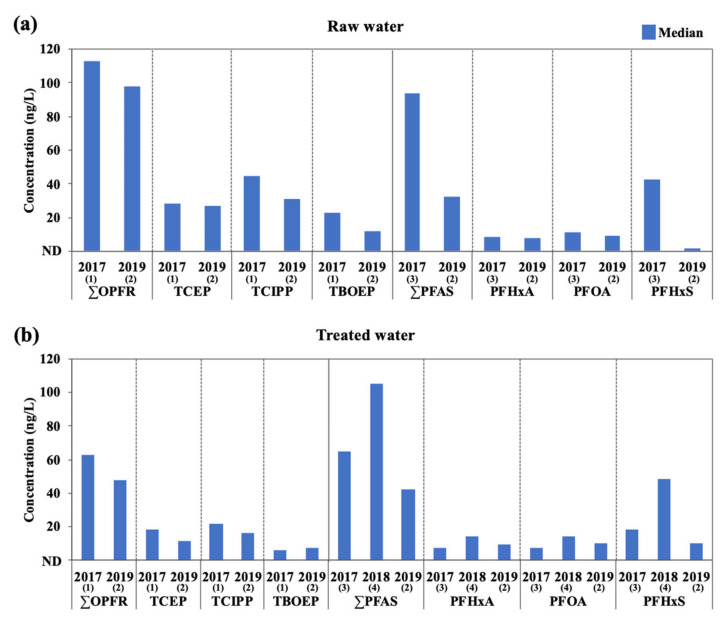
Temporal changes in concentrations of OPFR and PFAS in drinking water treatment plants (DWTPs) along the Nakdong River between 2017 and 2019: (**a**) raw water; (**b**) treated water. The data were taken from (1) Choo et al. (2020) [27], (2) this study, (3) Kim et al. (2020) [28], and (4) MOE (2018) [37].

**Figure 3 ijerph-18-02645-f003:**
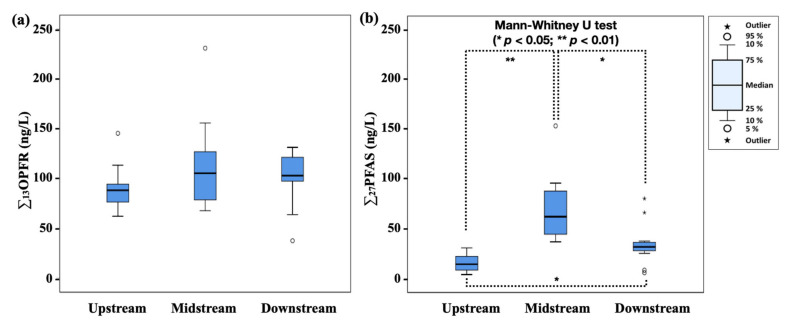
Comparison of ∑_13_OPFR and ∑_27_PFAS in raw water according to locations of water supplies: (**a**) ∑_13_OPFR; (**b**) ∑_27_PFAS. The sites were divided into the upstream (DWTPs 1–5), midstream (DWTPs 6–10), and downstream (DWTPs 11–18).

**Figure 4 ijerph-18-02645-f004:**
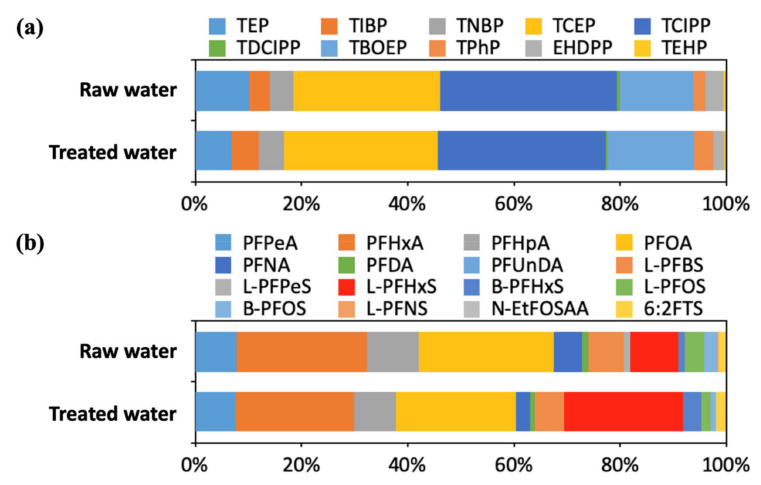
Proportions of OPFR and PFAS in DWTPs: (**a**) OPFR; (**b**) PFAS.

**Table 1 ijerph-18-02645-t001:** Removal tendencies of OPFR and PFAS in DWTPs.

Compound	*n*	Number of Cases According to Removal Efficiency Range					
		≥80%	<80% ≥50%	<50% ≥0%	<0% ≥−100%	<−100% ≥−200%	<−200%
TEP	35	12	11	10	1	1	0
TIBP	32	1	10	12	6	0	3
TNBP	36	1	15	14	5	0	1
TCEP	36	0	17	15	2	1	1
TCIPP	36	1	19	14	2	0	0
TDCIPP	16	12	0	1	1	1	1
TBOEP	36	0	13	17	5	0	1
TPhP	33	9	8	3	1	0	12
EHDPP	36	18	11	6	0	0	1
TEHP	19	0	8	10	0	0	1
∑_13_OPFR	36	0	18	16	1	1	0
PFPeA	28	4	0	1	18	0	5
PFHxA	36	0	0	7	29	0	0
PFHpA	36	2	0	11	23	0	0
PFOA	36	0	0	17	18	0	1
PFNA	36	11	0	10	14	1	0
PFDA	20	5	1	3	4	0	7
PFUnDA	4	4	0	0	0	0	0
∑_10_PFCAs	36	0	0	15	21	0	0
L-PFBS	35	1	0	6	18	0	0
L-PFPeS	20	18	0	0	2	0	0
L-PFHxS	34	1	0	3	7	4	19
B-PFHxS	29	1	0	4	4	3	17
L-PFOS	34	11	0	10	11	1	1
B-PFOS	30	13	1	9	7	0	0
L-PFNS	4	4	0	0	0	0	0
∑_9_PFSAs	36	0	2	6	12	7	9
N-EtFOSAA	3	0	0	0	0	0	3
6:2FTS	9	2	0	2	0	0	5
∑_5_Precursors	10	2	0	1	1	0	6
∑_27_PFAS	36	0	0	12	18	6	0

Removal rate (%) = (Concentration in raw water−Concentration in treated water) ÷ Concentration in raw water × 100.

## Data Availability

All of the data is presented in the article and associated Appendix A.

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
