# Peer review of "Organophosphate Flame Retardants and Perfluoroalkyl Substances in Drinking Water Treatment Plants from Korea: Occurrence and Human Exposure"

_ijerph, 2021, doi:10.3390/ijerph18052645_

Round 1
Reviewer 1 Report
IJERPH-1114263
This article is investigating the occurrence, concentrations, and human risk assessment of organophosphate flame retardants (OPFRs) and perfluoroalkyl substances (PFASs) in two types of water samples collected from Korean drinking water treatment plants (DWTPs). The environmental occurrence of OPFRs and PFASs is very important and especially linked with drinking water issues. The authors measured lots of OPFR and PFAS compounds in a correct way including acceptable QA/QC data. This article well formatted for this journal with clear writing in the whole manuscript. I recommend that this paper published in this journal after minor revision for publication.
Page 6 Lines 223-234.
The decreasing pattern of chemical concentration is not visible because standard deviation is varied. So it need to revise or pointing out the decreasing trends of OPFRs and PFASs were showed between sampling year.
Actually, the decreasing of PFHxS levels between the sampling years seems to have a major effect on the overall PFASs level, therefore it need to be emphasized in the manuscript.
Page 7 Lines 239-252.
As the authors mentioned, if there is a point source of PFAS such as WWTPs in the midstream, it cannot be explained that there is no change in the concentration of OPFRs. This is because WWTPs is known as a major source of contamination of OPFRs in surface water.
It would be better not to classify the sampling sites as up-, mid-, and down-stream, but to find the pollutant source by sampling sites and explain the difference in OPFRs and PFASs level.
Page 8 Lines 257-272.
What is the meaning of negative values of removal rates? It means that PFASs is input during DWTP process? Please clearly explain in the manuscript.
Page 8 Figure 4.
All the contents of the manuscript are expressed as treatment water, but in this figure, it is indicated as drinking water. Readers are concerned about confusion in their reading, so how about switch to treated water in the figure 4?
Reviewer 2 Report
This study reports the levels of OPFR and PFAS in water samples collected from before and after the drinking water treatment processes. Although study area is limited in certain area in Korea, the results of this study could be referred as a baseline information on the exposure to OPFR and PFAS through drinking water consumption. I think therefore this manuscript can be accepted for publication after revision. Please refer to the specific comments.
Line 24
Usually risk assessment is conducted using both median and P95 population. Can you include median values?
Line 26
I think you still cannot conclude if the levels are “acceptable” based only on this result, just suggesting non-cancer risk could be “low” or “less than known hazardous levels”.
Line 117, 124, 126, 132
What were included in the internal standards and syringe standards? Please specify the compounds you used.
Line 153-
Can you include the results of CRM (NIST SRM or other certified reference materials) in this section?
Line 175
Because you did not measure the daily intake directly, you can use “Estimated Daily Intake, EDI” for your calculation.
Line 189
You can cite and compare your data with the previous reports on drinking water samples from US, EU or Asian countries.
Line 193
Why don’t you impute the data distribution?
Line 230
“Lower when than” should be “lower than”.
Line 234
You can explain the accident briefly.
Figure 2
Min values for <LOD can be replaced to LOD value and should not be “zero”.
Line 248
OPFRs are also applied for textiles and electronics industries. Please discuss why OPFRs don’t follow this theory.
Figure 3
Sample number for each category should be indicated. You should describe details on how you categorize into 3 groups in M&M section.
